# JNC-1043, a Novel Podophyllotoxin Derivative, Exerts Anticancer Drug and Radiosensitizer Effects in Colorectal Cancer Cells

**DOI:** 10.3390/molecules27207008

**Published:** 2022-10-18

**Authors:** Jin-Hee Kwon, Na-Gyeong Lee, A-Ram Kang, In-Ho Ahn, In-Young Choi, Jie-Young Song, Sang-Gu Hwang, Hong-Duck Um, Jong-Ryoo Choi, Joon Kim, Jong Kuk Park

**Affiliations:** 1Division of Radiation Biomedical Research, Korea Institute of Radiological and Medical Sciences, Seoul 01812, Korea; 2Division of Life Sciences, Korea University, Seoul 02841, Korea; 3J&C Sciences Co., Ltd., KAIST Moonji Campus F712, 193 Moonji-ro, Yusung-Gu, Daejeon 305-732, Korea

**Keywords:** JNC-1043, radiosensitizer, topoisomerase inhibitor, ROS, apoptosis, colorectal cancer

## Abstract

The objective of this study was to determine whether (5S)-5-(4-benzyloxy-3,5-dimethoxy-phenyl)-5,9-dihydro-8H-furo [3’,4’:6,7] naphtho [2,3-d] [1,3]dioxol-6-one (JNC-1043), which is a novel chemical derivative of β-apopicropodophyllin, acts as a novel potential anticancer reagent and radiosensitizer in colorectal cancer (CRC) cells. Firstly, we used MTT assays to assess whether JNC-1043 could inhibit the cell proliferation of HCT116 and DLD-1 cells. The IC50 values of these cell lines were calculated as 114.5 and 157 nM, respectively, at 72 h of treatment. Using doses approximating the IC50 values, we tested whether JNC-1043 had a radiosensitizing effect in the CRC cell lines. Clonogenic assays revealed that the dose-enhancement ratios (DER) of HCT116 and DLD-1 cells were 1.53 and 1.25, respectively. Cell-counting assays showed that the combination of JNC-1043 and γ-ionizing radiation (IR) enhanced cell death. Treatment with JNC-1043 or IR alone induced cell death by 50~60%, whereas the combination of JNC-1043 and IR increased this cell death by more than 20~30%. Annexin V-propidium iodide assays showed that the combination of JNC-1043 and IR increased apoptosis by more 30~40% compared to that induced by JNC-1043 or IR alone. DCFDA- and MitoSOX-based assays revealed that mitochondrial ROS production was enhanced by the combination of JNC-1043 and IR. Finally, we found that suppression of ROS by N-acetylcysteine (NAC) blocked the apoptotic cell death induced by the combination of JNC-1043 and IR. The xenograft model also indicated that the combination of JNC-1043 and IR increased apoptotic cell death in tumor mass. These results collectively suggest that JNC-1043 acts as a radiosensitizer and exerts anticancer effects against CRC cells by promoting apoptosis mediated by mitochondrial ROS.

## 1. Introduction

Colorectal cancer (CRC), which is also known as bowel cancer, colon cancer, or rectal cancer, comprises cancer developed from the colon or rectum. CRC is the second most common cause of cancer death in the United States [1,2]. Between 2008 and 2017, there was a rapid decline in CRC incidence among those 65 years old and older, but the annual death rate from CRC increased by 1.3% each year in those younger than 50 years old [3]. The treatment modalities currently utilized against CRC are surgery, radiation therapy, chemotherapy, and immunotherapy [4,5]. The conventional chemo- or radio-therapies for CRC involve delivering drugs systemically or IR locally to the cancer site of patients. These modalities exert their effects by disrupting the cell cycle or inducing cell death. However, repeated usages of these anticancer treatments are usually accompanied by increased possibilities of resistance and/or side effects. Resistance can be intrinsic or acquired, and is believed to cause treatment failure in over 90% of resistant patients due to the induction of recurrence or metastasis, usually resulting in death [6,7]. To overcome these issues, researchers worldwide are seeking to develop various improved anticancer drugs and/or radiosensitizer candidates [8].

Podophyllotoxin (PPT) is a major material used in the development of novel anticancer drugs. PPT is an active plant ingredient that can be isolated as an abundant lignan from podophyllin, which is a type of resin produced by *Podophyllum peltatum Linnaeus*. PPT is known to exert immunosuppressive activity and antiviral effects against herpes, measles, influenza, and venereal warts, and has also been shown to act against skin cancer [9]. Given the beneficial pharmacological properties of PPT, many investigators have synthesized and characterized various derivatives with the aim of improving the biological activities and studying the underlying physiological mechanisms. Three representative semi-synthetic epipodophyllotoxin derivatives have been identified as anticancer drugs: etoposide, teniposide, and etopophos. Their anticancer activities are based on their ability to destroy the DNA replication machinery by binding DNA topoisomerases, which are ubiquitous enzymes that control the topological state of DNA in cells. Treatment of cells with these drugs leads to the formation of DNA-drug-enzyme complexes that facilitate the breakage of one or both of the DNA strands, eventually leading to cell death or apoptosis [10,11,12]. We previously isolated podophyllotoxin acetate (PA), a naturally occurring derivative of PPT, and presented it as a novel candidate anticancer drug, chemosensitizer, and radiosensitizer against non-small cell lung cancer (NSCLC) [13,14]. We also synthesized β-apopicropodophyllin (APP) from PPT and demonstrated that it is a potential anticancer drug and radiosensitizer. We showed that APP damages microtubule polymerization and DNA, thereby inducing cell cycle arrest and pro-apoptotic ER stress [15,16]. The radiosensitizing effect of APP resulted from the ROS accumulation-induced promotion of apoptosis. In the present study, we synthesized a novel PPT derivative from APP, (5*S*)-5-(4-benzyloxy-3,5-dimethoxy-phenyl)-5,9-dihydro-8*H*-furo [3’,4’:6,7] naphtho [2,3-*d*][1,3]dioxol-6-one (JNC-1043), and explored its anticancer drug and radiosensitizer effects and the underlying molecular mechanisms in CRC cells [17].

## 2. Results

### 2.1. JNC-1043 Inhibits Proliferation in CRC Cell Lines by Inducing DNA Damage

JNC-1043 was synthesized from APP (Figure 1a). Structural investigations using Nuclear Magnetic Resonance (NMR) as well as literature searches confirmed that the synthesized compound contains a novel chemical structure that enabled us to register a Korean Patent (No. 10-2090554) and apply for a U.S. Patent (U.S. Application Serial No. 16991190). We compared the anticancer effects of PA, APP, and JNC-1043 against the CRC cell lines, HCT116 and DLD-1, and the human colon fibroblast cell line, CCD-18CO (Figure 1b). Cells were treated with 100 nM PA, APP, and JNC-1043 for 72 h, and cell viability was determined with the MTT assay. Our results clearly showed that PA and APP exerted very similar percentages of cytotoxicity on CCD-18CO cells and the CRC cell lines, whereas JNC-1043 showed very similar levels of cytotoxicity against the CRC cell lines, but less against the CCD-18CO cells. More specifically, JNC-1043-treated CCD-18CO cells showed about ~30% less cell death than those treated with APP or PA. These results implied that JNC-1043 could selectively induce cancer cell death, exhibiting less cytotoxicity against normal cells.

To investigate the anticancer effect of JNC-1043, we examined its IC50 (50% inhibition of viability) values. HCT116 and DLD-1 cells were treated with various concentrations of JNC-1043 for 48 or 72 h, and MTT assays were performed. The IC50 values of JNC-1043 against HCT116 and DLD-1 cells at 48 or 72 h were 171.9 nM and 159.8 nM or 114.5 nM and 157 nM, respectively (Figure 2a,b, Table 1). The relatively low IC50 values of JNC-1043 supported its potential value as a candidate of cancer treatment drug.

To begin examining the anticancer mechanism of JNC-1043, we used an immunoblot assay to assess the phosphorylation level of H2AX (γH2AX) (Figure 2c). Our results revealed that JNC-1043 treatment of HCT116 and DLD-1 cells increased their levels of γH2AX, indicating that the anticancer effect of JNC-1043 might involve the induction of DNA damage. Since HCT116 has wild-type p53 and DLD-1 has a mutant p53 [18], these results also suggest that the anticancer effect of JNC-1043 is not affected by the genetic state of p53.

### 2.2. JNC-1043 Acts as a Radiosensitizer by Inhibiting Cell Growth In Vitro

To investigate whether JNC-1043 has a radiosensitizing effect on CRC cells, clonogenic, cell counting, and immunoblot assays were performed. Firstly, clonogenic assays were performed by pre-treating cells with JNC-1043 (110 nM JNC-1043 for HCT116 cells and 150 nM JNC-1043 for DLD-1 cells) for 24 h and then irradiating them with 1, 2, 3, or 4 Gy IR. The DER values were calculated as 1.528 for HCT116 cells and 1.25 for DLD-1 cells (Figure 3a, Table 2). These results indicate JNC-1043 might be a radiosensitizer candidate. A cell counting assay was performed on cells co-treated with 110 or 150 nM JNC-1043, respectively, and 3Gy IR. Our results revealed that JNC-1043 enhanced the growth-inhibiting effect of IR by more than two-fold (Figure 3b). Additionally, our immunoblot assay revealed that the γH2AX level was increased by ~50% in cells co-treated with JNC-1043 and IR, compared to the JNC-1043-only or IR-only groups (Figure 3c). Taken together, these results indicate that JNC-1043 might act as a novel radiosensitizer via amplifying DNA damage.

### 2.3. Combination of JNC-1043 and IR Enhances Apoptosis and DNA Damage

To explore the intracellular mechanism underlying the radiosensitizing effect of JNC-1043, we performed Annexin V-PI staining for apoptosis detection in HCT116 and DLD-1 cells. Apoptotic cell death was increased by ~20% in both CRC cell lines following co-treatment with JNC-1043 (at 110 or 150 nM, respectively) and IR compared to the levels seen in the JNC-1043-only and IR-only groups (Figure 4a). Immunoblot assays for the apoptosis-related proteins, Bcl-2, Bcl-X_L_, cleaved caspase-3, cleaved caspase-9, and cleaved PARP, were performed to assess whether the apoptotic cell death mechanism is increased in CRC cells co-treated with JNC-1043 and IR. Indeed, the levels of Bcl-2 and Bcl-X_L_ were decreased while those of cleaved caspase-3, caspase-9, and PARP were increased in cells co-treated with JNC-1043 and IR, relative to those in cells treated with JNC-1043 alone or IR alone (Figure 4b,b’). The release of cytochrome c from mitochondria to the cytosol was also increased in cells co-treated with JNC-1043 and IR (Figure 4c). These results indicate that a combination of JNC-1043 and IR can enhance the apoptosis of CRC cells by activating the apoptotic machinery.

### 2.4. The Combination of JNC-1043 and IR Induces Cell Cycle Arrest at G2/M Phase

Based on previous reports, we next explored whether JNC-1043 also could facilitate IR-induced cell cycle arrest. Cell cycle analysis was performed with HCT116 and DLD-1 cells co-treated with 110 (HCT-116) or 150 nM (DLD-1) of JNC-1043, respectively, and 1 Gy of IR for 4 (HCT-116) or 2 (DLD-1) h, respectively. After undergoing treatment with IR and JNC-1043, samples were fixed for 24 h, stained with PI/RNase staining buffer, and analyzed with a FACSort flow cytometer. Our results revealed that the population of CRC cells at the G2/M phase was increased by ~20% in samples co-treated with 1 Gy and 110 or 150 nM JNC-1043, respectively, compared to the level in the JNC-1043-only- treated and IR-only-treated groups (Figure 5a, Table 3). Immunoblot analyses showed that the expression levels of p21 and cyclin B1 were increased in cells co-treated with JNC-1043 and IR (Figure 5b). These results indicate that JNC-1043 can also induce cell cycle arrest at the G2 phase, which might eventually lead to cell death.

### 2.5. Co-Treatment of JNC-1043 and IR Enhances the Accumulation of ROS in CRC Cells

Since cancer cells are more vulnerable than normal cells to disruption of the redox balance and mitochondrial function, oxidative stress triggered by disruption in ROS homeostasis is an important principle of many cancer therapies, including chemotherapy, radiotherapy, and combination therapy [19,20]. Indeed, we previously showed that the disruption of ROS homeostasis is the main cause of radiosensitizer-induced apoptosis in our experimental settings [21,22].

Thus, intracellular and mitochondrial ROS detection assays were performed to determine whether JNC-1043 and IR co-treatment could enhance ROS production. The results of an intracellular ROS detection assay performed using H2DCFDA showed that the combination of JNC-1043 and IR enhanced the production of ROS by more than 20~40% compared to the levels seen in the JNC-1043-only or IR-only groups (Figure 6a). A mitochondrial ROS detection assay performed using Mitosox also showed that the combination of JNC-1043 and IR promoted the production of mitochondrial ROS by more than 30~40% compared to the levels seen in the JNC-1043-only or IR-only groups (Figure 6b).

Next, we sought to identify the source of the accumulating ROS by using JC-1 staining to detect mitochondrial membrane potential (Δ*Ψ**m*) (Figure 7a). Our results revealed that the combination of JNC-1043 and IR increased the mitochondrial potential by more than ~30% compared to the levels seen in the JNC-1043-only or IR-only groups. We also detected increased levels of cytochrome c into cytoplasm (Figure 7b). Treatment with the ROS scavenger, N-acetyl cysteine (NAC), abolished the increases in both intracellular and mitochondrial ROS production induced by co-treatment with JNC-1043 and IR (Figure 6c,d). Additionally, we detected that NAC treatment also decreased the increase in the mitochondrial potential and the levels of cytochrome c in cytoplasmic fractions (Figure 7a,b). These results suggest that the combination of JNC-1043 and IR may enhance ROS production by targeting mitochondria and affecting the homeostasis of ROS level.

### 2.6. ROS Is Important for the Radiosensitizing Effect of JNC-1043 against CRC Cells

As reported above, the combination of JNC-1043 and IR increased ROS production and apoptotic cell death. Therefore, we hypothesized that the increase in ROS could be linked to cell death in the radiosensitizing effect of JNC-1043 and tested this using NAC. Cell counting and Annexin V-PI staining assays were performed to determine whether pre-treatment with NAC could affect the disruption of ROS homeostasis by the combination of JNC-1043 and IR (Figure 8a,b). Indeed, pre-treatment with NAC reduced the cell number decrease and apoptotic cell death increase caused by co-treatment with JNC-1043 and IR by more than ~30% compared to cells that received the co-treatment without NAC. Pre-treatment with NAC also decreased the expression levels of Bcl-2/Bcl-X_L_, the levels of cleaved caspases and PARP (Figure 8c,c’), and the level of γH2AX (Figure 8d). Taken together, these results indicate that the combination of JNC-1043 and IR could enhance apoptotic cell death and DNA damage via mitochondrial membrane potential (Δ*Ψ**m*) disruption followed by mitochondrial ROS accumulation.

### 2.7. In Vivo Radiosensitization Effect of JNC-1043

Next, we tested whether the radiosensitization effect of JNC-1043 could be observed in an in vivo animal model. Xenograft model mice were prepared using HCT116 cells, and apoptotic cell death in xenografts was measured under various experimental conditions: mock control, IR only, JNC-1043 only, and the combination of JNC-1043/IR (Figure 9a). TUNEL assays of xenograft tissues were performed to determine the number of apoptotic cells. Quantitative analysis of TUNEL-stained areas showed that apoptotic cell death was increased by ~40% in the tumors of mice treated with the combination of JNC-1043 and IR.

Our results demonstrate that JNC-1043 has a radiosensitization effect against CRC cells via the induction of apoptosis, both in vitro and in an in vivo system. From our findings, we conclude that the combination of JNC-1043 and IR enhances apoptotic cell death via increases in mitochondrial ROS and DNA damage, followed by caspase activation. (Figure 9b).

## 3. Discussion

Radiosensitizers in cancer therapy are various materials that, when combined with IR, achieve greater tumor treatment effects than seen following the application of IR or the material alone. Radiosensitizers were historically characterized as: (1) suppressors of intracellular thiols or other endogenous radioprotective substances; (2) inducers of cytotoxic substances by intracellular radiolysis of the radiosensitizer; (3) inhibitors of repair mechanisms/molecules of biomolecules; (4) DNA-incorporating thymine analogs; and (5) oxygen mimics containing electrophilic activity [23]. Recently, given that in-depth research has revealed intracellular molecular mechanisms and led to the development of several novel materials, this classification was modified to focus on the structures of radiosensitizers. They are now classified into three categories: small molecules, macromolecules, and nanomaterials [24]. In a previous study, we examined the radiosensitizing small molecule, APP. First, we showed that the anticancer effects of APP were due to microtubule polymerization interruption and DNA damage, leading to cell cycle arrest accompanied by the induction of the pro-apoptotic ER stress-signaling pathway [15,16]. Additional studies showed that APP could function as a radiosensitizer against NSCLC and CRC cells [17]. Based on these reports, coupled with our observation that APP showed low toxicity and poor pharmacokinetic (PK) values in vivo (data not shown), we set out to improve APP as a potential anticancer agent. As shown in Figure 1, we newly synthesized JNC-1043 from APP and then validated that it has enhanced anticancer effects against CRC cells and less toxicity in normal cells. As shown in Figure 2C, JNC-1043 also exerted its anticancer effects on CRC cells by damaging DNA as evidenced by the increased phosphorylation of γH2AX.

When IR induces DNA damage to eukaryotic cells, cell death is triggered via the formation of DNA double-strand breaks (DSBs). DSB formation is immediately followed by histone H2AX phosphorylation (γH2AX) and the accumulation of numerous repair factors [25]. The accumulation of repair factors at the DSB region results in the formation of large irradiation-induced foci (IRIFs) that contain two major repair systems: homologous recombination (HR) and nonhomologous end joining (NHEJ) [26]. The major DSB-repair molecules for HR are members of the Rad52 epistasis group, whereas those for NHEJ are Mre11/Rad50/Nbs1 and Ku70/80/DNa-PK/Ligase IV [27]. The previous findings and our present data suggest that JNC-1043 could act intracellularly to initiate DNA damage and, thereby, function as a radiosensitizer. Indeed, JNC-1043 showed a radiosensitizing effect on CRC cells, suppressing cell viability by more than two-fold and increasing the γH2AX level by more than three-fold (Figure 3). Combination treatment with JNC-1043 and IR further enhanced apoptosis in vitro and in vivo (Figure 4 and Figure 8a and induced arrest at the G2 phase of the cell cycle (Figure 5). As shown in Figure 4, the combination of JNC-1043 and IR down-regulated anti-apoptotic Bcl-2 and Bcl-X_L_, up-regulated cleaved caspase-3, caspase-9, and PARP, and enhance the release of cytochrome c. It is known that released cytochrome c can form a complex with procaspase 9 and apoptosis protease-activating factor 1 (APAF1), which can activate caspase 9. Caspase 9 then activates pro-caspase 3 and pro-caspase 7, resulting in cell death [28,29,30]. The progression of cells from the G2 phase to the M phase is triggered by activation of the cyclin B1-dependent kinase, Cdc2, during a normal cell cycle [31,32,33]. Suppression of cyclin B1/Cdc2 activity would tend to cause cell cycle arrest at G2 phase, whereas a cell with elevated cyclin B1/Cdc2 activity would tend to enter mitosis [34]. Our data inducing p21 and cyclin B1 coincide with previous studies that described the induction of the G2/M phase cell cycle arrest following treatment of cells with microtubule inhibitors or IR has been observed with increasing cyclin B1 and p21 expression [35,36,37]. We also found that combination treatment with JNC-1043 and IR led to an increase in intracellular ROS. It is well known that exposure to penetrating IR can directly or indirectly lead to cellular stress via ROS generation [19,38]: ROS accumulation typically disrupts ROS homeostasis and, thereby, leads to oxidative stress, which can transiently or permanently injure cellular components such as proteins, lipids, RNA, and DNA [39]. ROS can induce almost all forms of DNA damage, including base modifications, strand breakage, and DNA cross-linking, all of which are known to be associated with cell death. Here, we show that co-treatment with JNC-1043 and IR enhanced the intracellular/mitochondrial ROS levels of CRC cells to induce apoptotic cell death (Figure 8). Pre-treatment with the ROS scavenger, N-acetylcysteine (NAC), was found to inhibit co-treatment-induced ROS production, cytochrome c release, H2AX phosphorylation, and apoptotic cell death (Figure 6, Figure 7 and Figure 8). These results suggest that increasing ROS under co-treatment with JNC-1043 and IR might be one of the most important effects of JNC-1043 as a radiosensitizer against CRC.

## 4. Materials and Methods

### 4.1. Cell Culture and Chemical Reagents

The HCT116, DLD-1, and CCD-18Co cell lines were purchased from the American Type Culture Collection (Rockville, MD, USA). All three cell lines were cultured in RPMI medium containing 10% fetal bovine serum and 1% penicillin, at 37 °C in a 5% CO_2_ incubator. For harvesting, the cells were washed with cold PBS and scraped into lysis buffer. JNC-1043 was synthesized by J&C Sciences (Daejeon, South Korea) as follows: a 60% dispersion of NaH in mineral oil (1.26 g, 31.4 mmol) was added portion-wise to a solution of 4′-demethyl-β-apopicropodophyllin (10 g, 26.2 mmol) in anhydrous DMF (100 mL) at 0 ℃. The mixture was stirred for 1 h at 0 ℃, benzyl bromide (6.7 g, 39.2 mmol) was added slowly, and stirring was continued for 12 h at ambient temperature. Water (300 mL) was added slowly to quench the reaction, and the mixture was extracted with ethyl acetate (2 × 100 mL). The combined organic layers were dried over MgSO_4_, filtered, and concentrated in vacuo. The residue was purified via column chromatography on SiO_2_ using ethyl acetate:n-hexane (1:4, *v/v*) to give JNC-1043 (7.7 g, 62%) as a white solid.

### 4.2. MTT Assay and IC50 Determination

HCT116, DLD-1, and CCD-18Co cells were seeded on 96-well plates (3 × 10^3^ cells/well) and exposed to various concentrations of JNC-1043 for 48 h or 72 h at 37 °C. MTT (3-(4,5-dimethylthiazol-2-yl)-2,5-diphenyltetrazolium bromide) solution (20 μL of 2 mg/mL) was added to each well, and the cells were incubated for 1 h at 37 °C. The formazan crystals generated in living cells were dissolved in 100 μL of DMSO, and the absorbances of individual wells were measured at 545 nm using a microplate reader (Molecular Devices, San Jose, CA, USA). The 50% inhibitory concentration (IC50) of JNC-1043 was calculated by performing a concentration–response analysis using the SoftMax Pro software Ver. 6.5 (Molecular Devices, Sunnyvale, CA, USA).

### 4.3. Clonogenic Assay

HCT116 and DLD-1 cells were seeded in 60-mm dishes at cell concentrations estimated to yield 20–100 colonies/dish (100, 200, 400, 600, 1000 cells/dish). After 24 h of incubation, the cells were treated with or without 110 nM and 150 nM JNC-1043, respectively, for 24 h, and then irradiated with ^137^Cs (as a source of γ-ray; Atomic Energy of Canada, Ltd., Mississauga, ON, Canada) at different doses (1, 2, 3, or 4 Gy). The cells were incubated for at least 14 days until colonies formed, and colonies larger than 200 μm in diameter were stained with 1% methylene blue in methanol. Stained colonies were counted using a colony counter (Imaging Products, Chantilly, VA, USA). The colony number of each dose was compensated with ratios that the maximum seeded cell number (1000 cells/dish) is divided by each seeded cell number. The dose-enhancement ratio (DER) value of each cell line was determined from the colony numbers using the Excel program (Microsoft Co., Redmond, WA, USA).

### 4.4. Cell Counting Assay

HCT116 and DLD-1 cells (1 × 10^5^ cells/60-mm dish) were incubated with or without 110 nM and 150 nM JNC-1043, respectively, and exposed to 3 Gy IR. The cells were incubated for 72 h at 37 °C, collected by trypsinization, and washed twice with cold PBS, and equal volumes of cells were stained with trypan blue. The cells were counted using an EVE^TM^ Automated Cell Counter (NanoEntek, Seoul, Korea). These experiments were repeated in triplicate.

### 4.5. Cell Cycle Analysis

HCT116 and DLD-1 cells (6 × 10^5^ cells/60-mm dish) were incubated with or without 110 nM and 150 nM JNC-1043, respectively, and exposed to 1 Gy IR for 2 or 4 h, respectively. The cells were collected by centrifugation at 1200 rpm for 1 min, washed twice with ice-cold phosphate-buffered saline (PBS), and fixed with ice-cold 70% ethanol. The fixed cells were stained with PI/RNase Staining Buffer (Becton Dickinson, Franklin Lakes, NJ, USA) for 15 min, and then analyzed with a FACSort flow cytometer (Becton-Dickinson, Franklin Lakes, NJ, USA). The percentages of cells in each phase of the cell cycle were determined using the Cell Quest software (Becton-Dickinson, Franklin Lakes, NJ, USA). These experiments were repeated in triplicate.

### 4.6. Immunoblot Analysis

HCT116 and DLD-1 cells were harvested and lysed with RIPA buffer (50 mM Tris-HCl, pH 7.6, 150 mM NaCl, 1% Triton X-100, 1% sodium deoxycholate, 0.1% SDS, 2 mM EDTA), and the lysates were centrifuged at 14,000× *g* for 20 min. The supernatants were removed, and the concentrations of proteins were measured at 280 nm using Bradford solution (Bio-Rad, Hercules, CA, USA) and a microplate reader (Molecular Devices). Twenty-milligram aliquots of proteins were heated at 95 °C for 1 min, resolved by SDS-PAGE, and transferred to a nitrocellulose membrane. After being blocked for 1 h at room temperature with 5% skim milk, the membranes were incubated at 4 °C with primary antibodies in 5% BSA solution. After 24 h of incubation, the membranes were washed and incubated with a secondary antibody for 1 h at room temperature. Bands were detected with the Clarity^TM^ Western ECL Substrate (Bio-Rad) and visualized using an Amersham ImageQuant 800 (GE Healthcare Bio-Sciences Corp., Marlborough, MA, USA). Primary antibodies were used against Bcl-2, Bcl-XL, pro-caspase-3, pro-caspase-9, cleaved caspase-3, cleaved caspase-9, pro-PARP, cleaved PARP, γ-H2AX (Cell Signaling Technology, Beverly, MA, USA), cyclin B1, and p21 (Santa Cruz Biotechnology, Santa Cruz, CA, USA). An anti-β-actin antibody (Sigma-Aldrich, St. Louis, MO, USA) was used as a loading control. The relative immunoblot band densities were determined via densitometry using the ImageJ software (NIH, Bethesda, MD, USA), normalized to that of each control, and analyzed with GraphPad Prism Ver. 5 (GraphPad Software, La Jolla, CA, USA). All experiments were repeated in triplicate.

### 4.7. Isolation of Mitochondrial and Cytosolic Fractions

HCT116 and DLD-1 cells were treated with 110 nM and 150 nM JNC-1043, respectively, irradiated 3 Gy IR, and incubated for 48 h at 37 °C. The cells were then incubated with trypsin-EDTA for 5 min, collected by centrifugation at 250 g for 1 min, resuspended with extraction buffer (1 M sucrose, 1 M HEPES, pH 7.4, 1 M KCl, 1 M MgCl_2_, 0.25 M EGTA, and 1 M DTT), homogenized, and centrifuged at 12,000× *g* at 4 °C for 15 min. The supernatants were used as the cytosolic fraction and the pellets were resuspended with lysis buffer (pH 7.6, 50 mM Tris-HCl, 150 mM NaCl, 1% Triton X-100, 1% sodium deoxycholate, 0.1% SDS, 2 mM EDTA) and used as the mitochondrial fraction. The fractions were immunoblotted with a cytochrome c antibody (Cell Signaling Technology, Beverly, MA, USA). VDAC (Sigma-Aldrich, St. Louis, MO, USA) was used as a loading control. These experiments were repeated in triplicate.

### 4.8. Annexin V-Propidium Iodide Assay

HCT116 and DLD-1 cells (1 × 10^5^ cells/60-mm dish) were incubated with or without 110 nM and 150 nM JNC-1043, respectively, exposed to 3 Gy IR, and incubated for 72 h. The cells were then collected by trypsinization, washed twice with cold PBS, and stained with an FITC Annexin V apoptosis detection kit I (Becton-Dickinson, Franklin Lakes, NJ, USA) as described by the manufacturer. A FACSort flow cytometer (Becton-Dickinson, Franklin Lakes, NJ, USA) was used to measure the fraction of apoptotic cells (*x*-axis: FL1 channel and *y*-axis: FL-2 channel). These experiments were repeated in triplicate.

### 4.9. Cellular ROS Detection Assay

HCT116 and DLD-1 cells (5 × 10^5^ cells/60-mm dish) were incubated with or without 110 nM and 150 nM JNC-1043, respectively, and exposed to 3 Gy IR. After incubation for 24 h, the cells were stained with 25 mM H2DCFDA (Merck, Darmstadt, Germany) for 5 min and trypsinized. The cells were harvested by centrifugation and resuspended with cold PBS. A FACSort flow cytometer (Becton-Dickinson, Franklin Lakes, NJ, USA) was used for the detection and analysis of intracellular ROS (*x*-axis, FL1 channel; *y*-axis, Counts). These experiments were repeated in triplicate.

### 4.10. Mitochondrial ROS Detection Assay

HCT116 and DLD-1 cells (4 × 10^5^ cells/60-mm dish) were incubated with or without 110 nM and 150 nM JNC-1043, respectively, and exposed to 3 Gy IR. After incubation for 48 h, the cells were treated with 5 µM MitoSOX (Molecular Probes/Invitrogen, Eugene, OR, USA) for 20 min and trypsinized. The cells were harvested by centrifugation and resuspended with ice-cold PBS. A FACSort flow cytometer (Becton-Dickinson, Franklin Lakes, NJ, USA) was used for the detection and analysis of mitochondrial ROS (*x*-axis, FL2 channel; *y*-axis, Counts). These experiments were repeated in triplicate.

### 4.11. Mitochondrial Membrane Potential (ΔΨm) Detection Assay

HCT116 and DLD-1 cells (4 × 10^5^ cells/60-mm dish) were incubated with or without 110 nM and 150 nM JNC-1043, respectively, and exposed to 3 Gy IR. After incubation for 48 h, the cells were treated with 10 µM JC-1 (Molecular Probes/Invitrogen, Eugene, OR, USA) for 20 min and trypsinized. The cells were harvested by centrifugation and resuspended with ice-cold PBS. A FACSort flow cytometer (Becton Dickinson) was used for the detection and analysis of mitochondrial membrane potential (*x*-axis, FL1 channel; *y*-axis, Counts). These experiments were repeated in triplicate.

### 4.12. TUNEL Assay

All animal experiments were performed using approved protocols of the Institutional Animal Care and Use Committee (KIRAMS 2021-0017). The in vivo radiosensitization effects of JNC-1043 were measured using a xenograft model constructed by subcutaneously injecting 1 × 10^7^ HCT116 cells/mouse into 6-week-old BALB/cAnNCrj-nu/nu mice (KOATECH, Seoul, Korea). The mice were divided into four groups (n = 3 mice/group): control (mock-treated), IR treatment only (IR only), JNC-1043 treatment only (JNC-1043 only), and co-treatment with JNC-1043 and IR (IR+JNC-1043). When the xenografts reached 100−120 mm^3^ in size, 40 mg/kg JNC-1043 in DMSO was intratumorally injected into mice of the JNC-1043-only and IR+JNC-1043 groups. The IR-only and control groups were intratumorally injected with an equal volume of DMSO (vehicle). At 6 h after the injection, mice in the IR-only and IR+JNC-1043 groups were locally irradiated with 1.5 Gy using a ^60^Co γ-ray source (Theratron 780; AECL Ltd., Mississauga, Ontario). Irradiation was repeated three times at 3- to 4-day intervals for a total of 12 days. Mice were sacrificed 30 days after the start of the experiment. Extracted tumors were fixed with formaldehyde, paraffin-embedded, and sliced. TUNEL assay for the detection of dUTP nicks was performed using Super Bio Chips (Seoul, South Korea). The TUNEL-positive cells were measured for each group, and the ratios of TUNEL-positive to total cells in each image were measured with the Image J software (NIH, Bethesda, MD, USA). Percentages were calculated relative to the value obtained from the control group.

### 4.13. Statistical Analysis

Data were analyzed using GraphPad Prism Ver. 5 (GraphPad Software, La Jolla, CA, USA). The significance of differences between experimental groups was determined using two-way ANOVA and Student’s t-test. *p*-values < 0.05 were considered significant, and the individual *p*-values are denoted in the figures by asterisks (*; *p* < 0.05, **; *p* < 0.01, ***; *p* < 0.001). The number above each point or bar represents the mean results from three independent experiments and the error bars signify the standard deviation (SD).

## 5. Conclusions

Taken together, our present findings reveal the anticancer and radiosensitizer effects of JNC-1043. Our detailed intracellular mechanistic studies and the lower toxicity of JNC-1043 against normal cells demonstrate that JNC-1043 might be a candidate radiosensitizer for inducing apoptotic cell death via DNA damage/cell cycle arrest/ROS-triggered apoptosis in CRC cells.

## Figures and Tables

**Figure 1 molecules-27-07008-f001:**
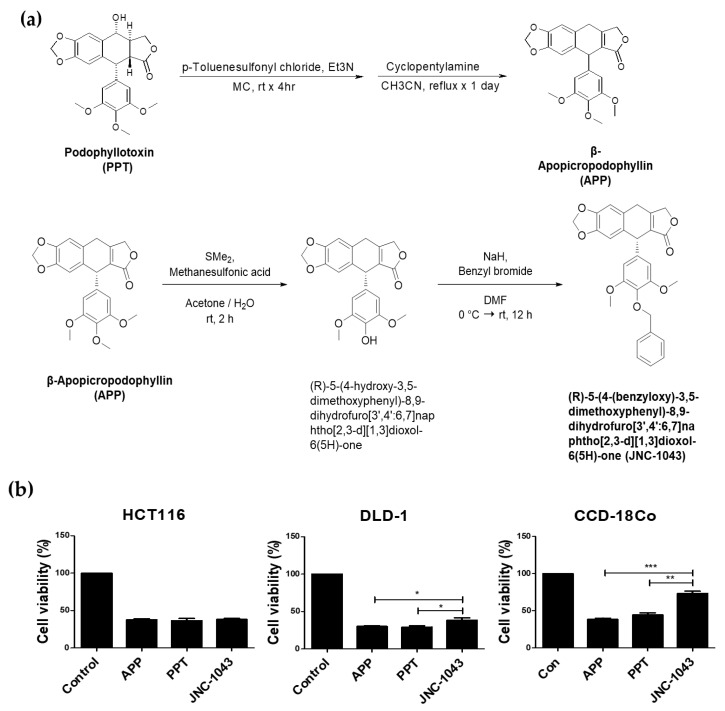
Synthesis of JNC-1043 and comparison of its toxicity. (**a**) Chemical synthesis of (5*S*)-5-(4-benzyloxy-3,5-dimethoxy-phenyl)-5,9-dihydro-8*H*-furo [3’,4’:6,7] naphtho [2,3-*d*][1,3]dioxol-6-one (JNC-1043). (**b**) The human colon fibroblast cell line, CCD-18CO, and the CRC cell lines, HCT116 and DLD-1, were treated with 100 nM APP, PPT, and JNC-1043 for 72 h, and cell viability was detected with the MTT assay. * *p* < 0.05, ** *p* < 0.01, *** *p* < 0.001.

**Figure 2 molecules-27-07008-f002:**
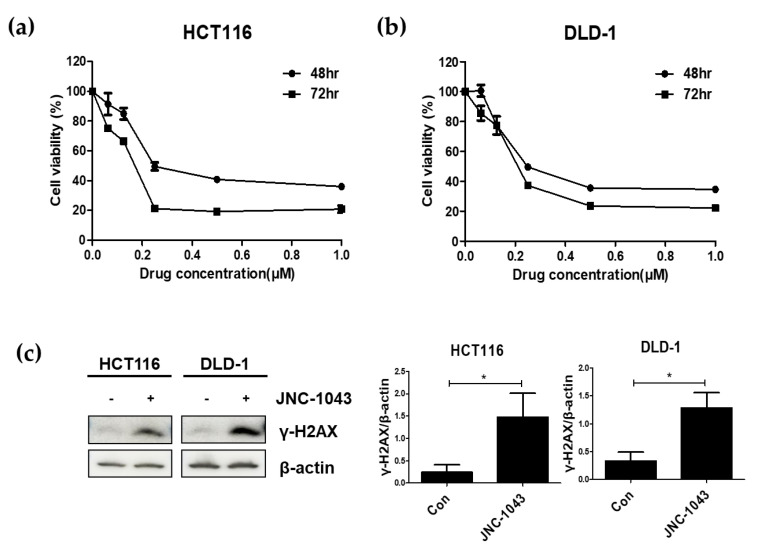
Anticancer effect of JNC-1043. The CRC cell lines, HCT116 (**a**) and DLD-1 (**b**), were treated with 62.5, 125, 250, 500, and 1000 nM JNC-1043 for 48 or 72 h, and cell viability was measured with the MTT assay. IC50 values for JNC-1043 against CRC cell lines were calculated as described in the Section 4. (**c**) Immunoblot assay for the detection of γH2AX activation in cells treated with JNC-1043. HCT116 and DLD-1 cells were incubated with 110 or 150 nM JNC-1043, respectively, and incubated for 24 h prior to harvesting. The relative band densities were analyzed as described in the Section 4. All experiments were repeated in triplicate, and the bands in the figures show representative data. * *p* < 0.05.

**Figure 3 molecules-27-07008-f003:**
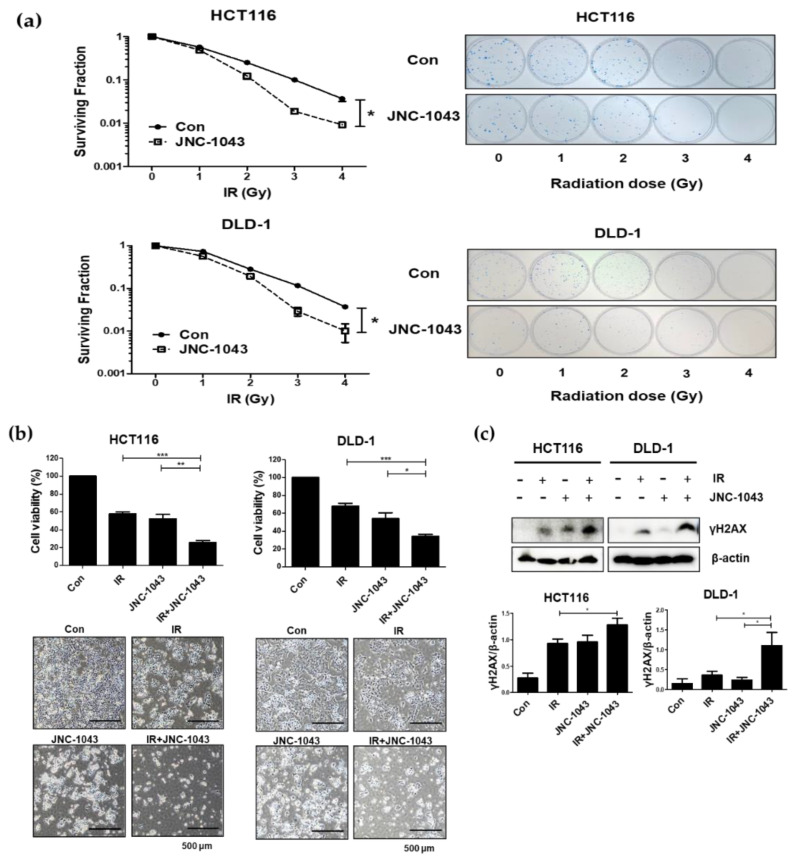
Co-treatment with JNC-1043 and IR suppresses cell growth. (**a**) Clonogenic assay to determine the radiosensitizing effect of JNC-1043 on HCT116 and DLD-1 cells. (**b**) Cell counting assay for detection of cell death. Cell counting assays were performed as described in the Materials and Methods. ‘Con’, DMSO-treated mock control; ‘IR’, treatment with 3 Gy IR only; ‘JNC-1043′, treatment with 110 or 150 nM (for HCT116 and DLD-1 cells, respectively) JNC-1043 only; ‘IR+JNC-1043′, combination of 3 Gy IR and 110 or 150 nM JNC-1043, respectively. The lower panel shows microscopic images obtained prior to cell detachment. Experiments were repeated in triplicate, and the presented results indicate the mean of triplicate assays. Each bar in the pictures indicates 500 μm. (**c**) Immunoblot assay for detection of γH2AX. HCT116 and DLD-1 cells were co-treated with 110 or 150 nM of JNC-1043, respectively, and 3 Gy IR for 24 h, and harvested. The relative band densities were analyzed as described in the Materials and Methods. All experiments were repeated in triplicate, and the bands in the figures show representative data. * *p* < 0.05, ** *p* < 0.01, *** *p* < 0.001.

**Figure 4 molecules-27-07008-f004:**
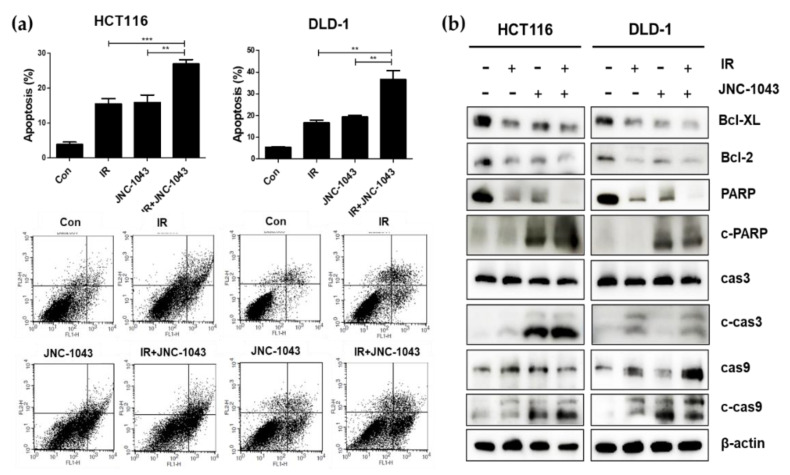
Co-treatment with JNC-1043 and IR induces apoptotic cell death. (**a**) Annexin V-PI assay for detection of apoptosis. HCT116 and DLD-1 cells were treated with 110 or 150 nM JNC-1043, respectively, and 3 Gy IR for 72 h. The assay method is described in the Materials and Methods. The lower panel shows representative FACSort flow cytometry images obtained prior to quantitative analyses. ‘Con’, DMSO-treated mock control; ‘IR’, treatment with 3 Gy IR only; ‘JNC-1043′, treatment with 110 or 150 nM JNC-1043 only; ‘IR+JNC-1043′, combination of 3 Gy IR and 110 or 150 nM JNC-1043. Experiments were repeated in triplicate, and the results indicate the mean of the triplicate assay. (**b**) Immunoblot assay for detecting Bcl-2, Bcl-XL, cleaved caspase-3 (cas-3), cleaved caspase-9 (cas-9), and cleaved PARP in cells treated with JNC-1043 and IR. ‘c-cas3′, ‘c-cas9′ and ‘c-PARP’ indicate the cleaved forms. HCT116 and DLD-1 cells (1 × 10^5^) were incubated with or without 110 or 150 nM JNC-1043, respectively, exposed to 3 Gy IR, and incubated for 72 h prior to harvest. Each graph under the panel presents the statistical analysis of the immunoblot bands. (**b’**) The relative band densities of (**b**) were analyzed as described in the Materials and Methods. (**c**) Immunoblot assay for detecting release of cytochrome c (Cyto-C) from mitochondria to the cytosol in cells co-treated with JNC-1043 and IR. HCT116 and DLD-1 cells (3 × 10^5^) were incubated with or without 110 or 150 nM JNC-1043, respectively, and exposed to 3 Gy IR for 48 h. Details of the assay methods are described in the Materials and Methods. Each graph in the right panel presents the statistical analysis of the immunoblot bands. All experiments were repeated in triplicate, and the bands in the figures show representative data. * *p* < 0.05, ** *p* < 0.01, *** *p* < 0.001.

**Figure 5 molecules-27-07008-f005:**
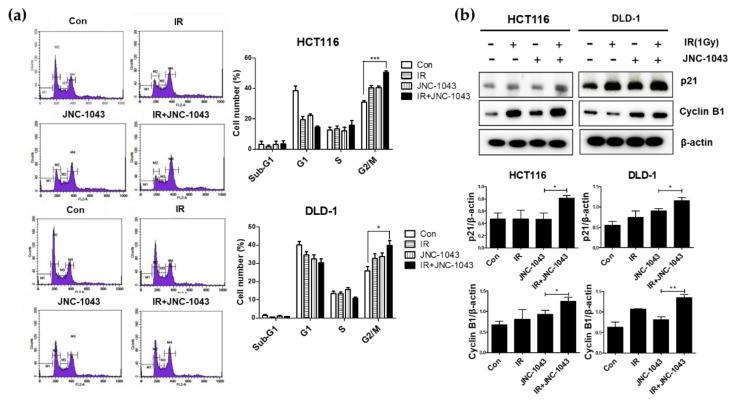
Co-treatment with JNC-1043 and IR promotes G2/M phase arrest in the cell cycle. (**a**) Cell cycle analyses. HCT116 and DLD-1 cells (8 × 10^5^) were treated with or without 110 or 150 nM JNC-1043, respectively, exposed to 1 Gy IR, and incubated for 2 or 4 h, respectively, prior to harvest. Cell cycle parameters of JNC-1043 and IR co-treated HCT116 and DLD-1 cells were analyzed with a FACSort flow cytometer. ‘Con’, DMSO-treated mock control; ‘IR’, treatment with 1 Gy IR only; ‘JNC-1043′, treatment with 110 or 150 nM JNC-1043 only; ‘IR+JNC-1043′, combination of 1 Gy IR and 110 nM or 150 nM JNC-1043. (**b**) Immunoblot assay for detection of cyclin B1, p21, and β-actin. HCT116 and DLD-1 cells (8 × 10^5^) were treated with or without 110 or 150 nM JNC-1043, respectively, exposed to 1 Gy IR, and incubated for 2 or 4 h, respectively, prior to harvest. Each graph in the right panel presents the statistical analysis of the immunoblot bands. The relative band densities were analyzed as described in the Materials and Methods. All experiments were repeated in triplicate, and the bands in the figures show representative data. * *p* < 0.05, ** *p* < 0.01, *** *p* < 0.001.

**Figure 6 molecules-27-07008-f006:**
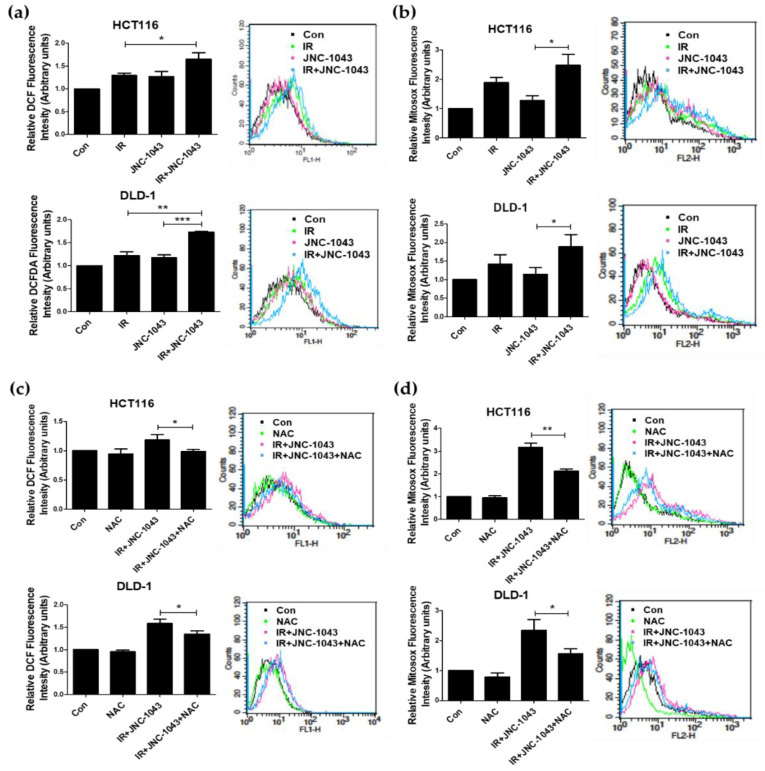
Co-treatment with JNC-1043 and IR leads to accumulation of mitochondrial ROS. (**a**,**b**) Detection of intracellular ROS (**a**), and mitochondrial ROS (**b**) using a FACSort flow cytometer. HCT116 and DLD-1 cells (5 × 10^5^) were treated with or without 110 or 150 nM JNC-1043, respectively, exposed to 3 Gy IR, and incubated for 24 h prior to harvest. ‘Con’, mock control; ‘IR’, treatment with 3 Gy IR only; ‘JNC-1043′, treatment with 110 nM or 150 nM JNC-1043 only; ‘IR+JNC-1043′, combination of 3 Gy IR and 110 nM or 150 nM JNC-1043. Experiments were repeated in triplicate, and the presented results indicate the mean of triplicate assays. (**c**,**d**) Inhibition of ROS production by NAC. HCT116 and DLD-1 cells (2.5 × 10^5^) were treated with or without 110 or 150 nM JNC-1043, respectively, exposed to 3 Gy IR, and incubated for 48 h prior to harvest. ‘Con’, mock control; ‘NAC’, HCT116 or DLD-1 cells pre-treated with 5 or 1 mM NAC, respectively; ‘IR+JNC-1043′, combination of 110 or 150 nM JNC-1043 with 3 Gy IR; ‘IR+JNC-1043+NAC’, combination of 3 Gy IR with 110 or 150 nM JNC in cells pre-treated with 5 or 1 mM NAC, respectively. Intracellular ROS (**c**) and mitochondrial ROS (**d**) were detected with a FACSort flow cytometer. Experiments were repeated in triplicate, and the results indicate the mean of triplicate assays. * *p* < 0.05, ** *p* < 0.01, *** *p* < 0.001.

**Figure 7 molecules-27-07008-f007:**
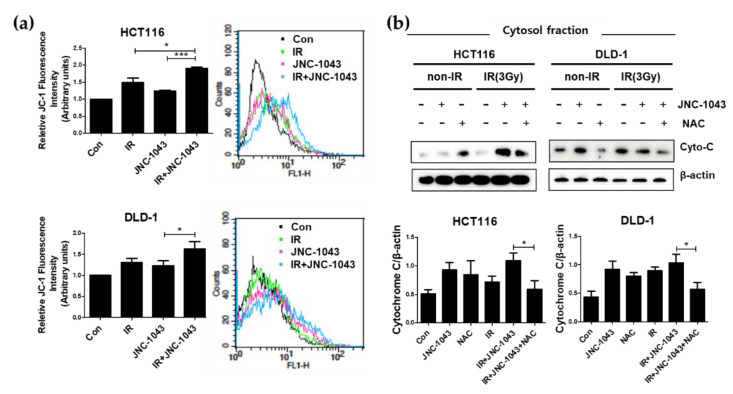
Co-treatment with JNC-1043 and IR disrupts mitochondrial potential. (**a**) HCT116 and DLD-1 cells (5 × 10^5^) were treated with or without 110 or 150 nM JNC-1043, exposed to 3 Gy IR and incubated for 24 h prior to harvest. ‘Con’, mock control; ‘IR’, treatment with 3Gy IR only; ‘JNC-1043′, treatment with 110nM or 150nM JNC-1043 only; ‘IR+JNC-1043′, combination of 3 Gy IR and110nM or 150nM JNC-1043. Mitochondrial potential in JNC-1043 and irradiation co-treated HCT116 and DLD-1 cells were detected with a FACSort flow cytometer. Experiments were repeated in triplicate, and the results indicated the mean of triplicate assays. (**b**) Immunoblot assay for detecting the release of cytochrome c from mitochondria to cytosol. JNC-1043 and IR co-treated CRC cells for 72 h with or without NAC pre-treatment for 2 or 1 h, respectively. HCT116 and DLD-1 cells (3 × 10^5^) were incubated with or without 110 or 150 nM JNC-1043 and exposed to 3 Gy IR for 48 h. Detail assay methods are described in the Material and Methods. Each graph under the panel indicates the statistical analysis of the immunoblot bands. The relative band densities were determined via densitometry using ImageJ software (NIH, Bethesda, MD, USA) and then normalized to that of each control. Experiments were repeated in triplicate, and bands in the figures show representative data. * *p* < 0.05, *** *p* < 0.001.

**Figure 8 molecules-27-07008-f008:**
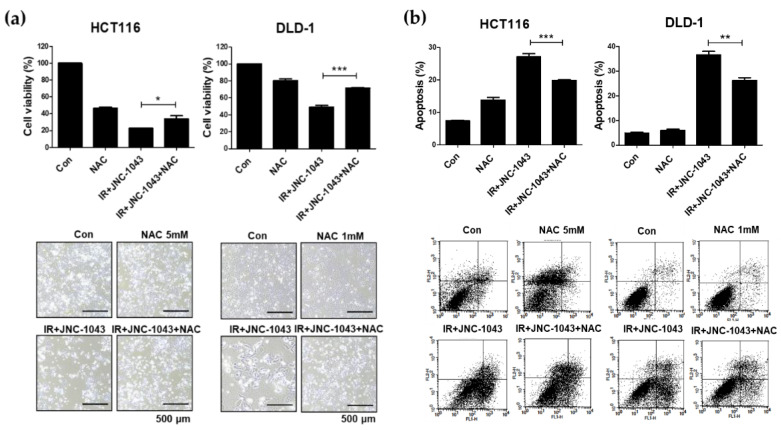
Blockage of ROS decreases the radiosensitizing ability of JNC-1043. (**a**) Cell viability in JNC-1043 and IR co-treated CRC cells with or without NAC pre-treatment for 2 or 1 h, respectively. HCT116 and DLD-1 cells (1 × 10^5^) were incubated with or without 110 or 150 nM JNC-1043, respectively, exposed to 3 Gy IR and incubated for 72 h prior to harvest. ‘Con’ DMSO-treated mock control; ‘NAC’, HCT116 or DLD-1 cells pre-treated with 5 or 1 mM NAC, respectively; ‘IR+JNC-1043′, combination of 110 or 150 nM JNC-1043 with 3 Gy IR; ‘IR+JNC-1043+NAC’, combination of 3 Gy IR with 110 or 150 nM JNC-1043 in cells pre-treated with 5 or 1 mM NAC, respectively. The lower panel shows microscopic images obtained prior to cell detachment. Each bar in the pictures indicates 500 μm. (**b**) Annexin V-PI uptake assay for detection of apoptosis in CRC cells co-treated with JNC-1043 and IR for 72 h with or without NAC pretreatment for 2 or 1 h, respectively, as detected using a FACSort flow cytometer. The lower panel shows representative FACSort flow cytometry images obtained prior to quantitative analyses. Experiments were repeated in triplicate, and the results indicate the mean of triplicate assay. (**c**,**d**) Immunoblot assay for detection of apoptosis-related proteins (**c**) and its relative band densities (**c’**), and γH2AX activation (**d**). Activation of caspase-3, caspase-9, PARP, and γH2AX were detected in cells pre-treated with NAC and treated with JNC-1043 and IR for 72 h. ‘cas3′, caspase-3; ‘cas9′, caspase-9; ‘c-cas3′, cleaved caspase-3; ‘c-cas9′, cleaved caspase-9; ‘c-PARP’, cleaved PARP. The relative band densities were analyzed as described in the Materials and Methods. All experiments were repeated in triplicate, and the bands in the figures show representative data. * *p* < 0.05, ** *p* < 0.01, *** *p* < 0.001.

**Figure 9 molecules-27-07008-f009:**
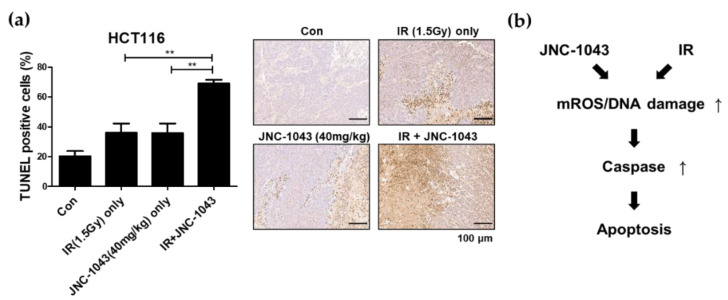
Combination treatment with JNC-1043 and IR inhibits tumor growth in vivo. (**a**) TUNEL assays measuring apoptotic cells in mouse xenografts. Mice injected with CRC cells were divided into four groups: control group, IR-only group (1.5 Gy), JNC-1043-only group (40 mg/kg), and IR+JNC-1043 group (in which mice were pretreated with 40 mg/kg JNC-1043 for 6 h prior to IR). Apoptotic cells in each xenograft were detected by TUNEL assay. The dark brown area in each tissue sample, indicating staining with the TUNEL reagent, was detected using Image J Ver.1.8.0. The graph indicates quantitative analyses of the ratios of TUNEL-stained apoptotic cells to total cells. ‘Con’, mock control; ‘IR only’, treatment with 1.5 Gy IR only; ‘JNC-1043 only’, treatment with 40 mg/kg JNC-1043 only; ‘IR+JNC-1043′, co-treatment with 1.5 Gy IR and 40 mg/kg JNC-1043. Each bar in the tissue images indicates 100 μm. Experiments were repeated in triplicate, and the results indicate the mean of triplicate assays. (**b**) Schematic diagram showing the radiosensitization effect of JNC-1043. ** *p* < 0.01.

**Table 1 molecules-27-07008-t001:** IC50 value.

	IC50 Value (µM)	
	48 h	72 h
HCT116	0.1719	0.1145
DLD-1	0.1598	0.1570

Note: IC50 (the 50% inhibitory concentration) was calculated from a concentration-response analysis performed with the SoftMax Pro software (Molecular Devices, Sunnyvale, CA, USA).

**Table 2 molecules-27-07008-t002:** DER analysis.

	DER (Dose-Enhancement Ratio)
	110 or 150 nM
HCT116	1.53
DLD-1	1.25

Note: DER (Dose-Enhancement Ratio) was calculated by clonogenic assay. Irradiation dose at 0.1 of survival fraction was used all the groups. For HCT116 cells, the doses were determined as 4.17 (control, irradiation only) and 2.72 (co-treatment with 110 nM JNC-1043 and irradiation). For DLD-1 cells, the doses were determined as 3.21 (control, irradiation only) and 2.57 (co-treatment with 150 nM JNC-1043 and irradiation). DER was calculated with these doses: irradiation dose at survival fraction of 0.1 IR only treatment/irradiation dose at survival fraction of 0.1 co-treatment with JNC-1043 group.

**Table 3 molecules-27-07008-t003:** Effect of JNC43 and irradiation on the cell cycle distribution of CRC cells.

HCT116	Con	IR (1 Gy)	JNC-1043	IR+JNC-1043
Sub-G1	3.2 ± 0.3	1.7 ± 0.1	3.1 ± 0.3	3.5 ± 0.3
G1	38.5 ± 5.9	19.5 ± 3.4	22.2 ± 2.3	14.4 ± 2.0
S	12.6 ± 0.9	13.3 ± 0.9	12.2 ± 1.9	15.8 ± 4.0
G2/M	31.1± 2.1	40.5 ± 3.1	40.7 ± 0.6	50.7 ± 2.3
**DLD-1**	**Con**	**IR (1 Gy)**	**JNC-1043**	**IR+JNC-1043**
Sub-G1	1.4 ± 0.3	0.5 ± 0.3	1.1 ± 0.1	0.8 ± 0.1
G1	40.1 ± 4.0	34.7 ± 3.5	32.4 ± 5.0	30.4 ± 4.0
S	13.4 ± 3.0	13.4 ± 1.9	15.7 ± 1.9	10.8 ± 1.5
G2/M	25.9 ± 5.4	32.8 ± 5.0	33.8 ± 3.1	40 ± 5.0

Note: Con, Control; IR (1 Gy), treatment with 1 Gy IR only; ‘JNC43’, treatment with 110 (for HCT116 cells) or 150 nM (for DLD-1 cells) JNC43 only; ‘IR+JNC43’, combination of 1 Gy IR and 110 or 150 nM JNC43.

## Data Availability

Not applicable.

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
