# Peer review of "JNC-1043, a Novel Podophyllotoxin Derivative, Exerts Anticancer Drug and Radiosensitizer Effects in Colorectal Cancer Cells"

_molecules, 2022, doi:10.3390/molecules27207008_

Round 1

Reviewer 1 Report

In this article, "JNC-1043, a Novel Podophyllotoxin Derivative, Exerts Anticancer Drug and Radiosensitizer Effects in Colorectal Cancer Cells," the authors examined the antitumor and radiosensitizing effects of JNC-1043, a novel podophyllotoxin derivative. Moreover, they proposed a molecular mechanism of action of JNC-1043 in CRC cells.

The work  is interesting. However, the following concerns should be addressed before the editor makes a decision on this work.

1.      Figure 1 shows only the JNC-1043 synthesis scheme. Cell viability results in the MTT assay were not included. The drawing should be corrected to match the caption and description of the results in the manuscript text.

2.       In Figure 4, the results described in the text of the manuscript and in the caption below the figure are missing. Figures 4a, 4b, b’ are not attached.

3.       Chapter 2.4, lines 194-196 and 198-200

Unclear description of the results.

Minor suggestions

In my opinion, it would be easier to analyze the work if the type of letters in the drawings and in captions under the drawings were the same (e.g. A or a, but the same in the drawings and in captions).

Author Response

The work  is interesting. However, the following concerns should be addressed before the editor makes a decision on this work.

  1. Figure 1 shows only the JNC-1043 synthesis scheme. Cell viability results in the MTT assay were not included. The drawing should be corrected to match the caption and description of the results in the manuscript text.

           -> We add MTT assay results and correct errors in Figure 1.

  1. In Figure 4, the results described in the text of the manuscript and in the caption below the figure are missing. Figures 4a, 4b, b’ are not attached.

        -> We add Figures 4a, 4b, b’.

  1. Chapter 2.4, lines 194-196 and 198-200

Unclear description of the results.

    -> We changed manuscript as follows:

       Lines 194-196:

Cell cycle analysis was performed with HCT116 and DLD-1 cells co-treated with 110 (HCT-116) or 150 nM (DLD-1) of JNC-1043, respectively, and 1 Gy of IR for 4 (HCT-116) or 2 (DLD-1) h, respectively.

      Lines 198-200:

Our results revealed that population of CRC cells at G2/M phase was increased by ~20% in samples co-treated with 1 Gy and 110 or 150 nM JNC-1043, respectively, compared to the level in in the JNC-1043 only- treated and IR only-treated groups.

Minor suggestions

In my opinion, it would be easier to analyze the work if the type of letters in the drawings and in captions under the drawings were the same (e.g. A or a, but the same in the drawings and in captions).

      -> We unify the type of letters in Figures.

Reviewer 2 Report

Manuscript ID molecules-1939818

Title: JNC-1043, a Novel Podophyllotoxin Derivative, Exerts Anti-cancer Drug and Radiosensitizer Effects in Colorectal Cancer Cells

The above-titled article reports the Radiosensitizer Effects of Podophyllotoxin Derivative (JNC-10430 in Colorectal Cancer Cells. The manuscript is well presented with appropriate experiments. But the same work and all the studies were already published in the public domain at “Transactions of the Korean Nuclear Society Spring Meeting” with the title: Novel podophyllotoxin derivative exerts effects of an anti-cancer drug and a radiosensitizer. The authors should comment on it.

Author Response

The above-titled article reports the Radiosensitizer Effects of Podophyllotoxin Derivative (JNC-10430 in Colorectal Cancer Cells. The manuscript is well presented with appropriate experiments. But the same work and all the studies were already published in the public domain at “Transactions of the Korean Nuclear Society Spring Meeting” with the title: Novel podophyllotoxin derivative exerts effects of an anti-cancer drug and a radiosensitizer. The authors should comment on it.

-> This public domain published contents was POSTER for Korean Nuclear Society Spring Annual Meeting.

-> Official Journal of Korean Nuclear Society is “Nuclear Engineering and Technology (https://www.journals.elsevier.com/nuclear-engineering-and-technology)”. We did NOT publish our present manuscript to Nuclear Engineering and Technology.

Round 2

Reviewer 2 Report

Accept